

# Suppression of *hesA* mutation on nitrogenase activity in *Paenibacillus polymyxa* WLY78 with the addition of high levels of molybdate or cystine

Xiaomeng Liu, Xiyun Zhao, Xiaohan Li and Sanfeng Chen

State Key Laboratory of Agrobiotechnology and College of Biological Sciences,
China Agricultural University, Beijing, China

## ABSTRACT

The diazotrophic *Paenibacillus polymyxa* WLY78 possesses a minimal nitrogen fixation gene cluster consisting of nine genes (*nifB nifH nifD nifK nifE nifN nifX hesA* and *nifV*). Notably, the *hesA* gene contained within the *nif* gene cluster is also found within *nif* gene clusters among diazotrophic cyanobacteria and *Frankia*. The predicted product HesA is a member of the ThiF-MoeB-HesA family containing an N-terminal nucleotide binding domain and a C-terminal MoeZ/MoeB-like domain. However, the function of *hesA* gene in nitrogen fixation is unknown. In this study, we demonstrate that the *hesA* mutation of *P. polymyxa* WLY78 leads to nearly complete loss of nitrogenase activity. The effect of the mutation can be partially suppressed by the addition of high levels of molybdate or cystine. However, the nitrogenase activity of the *hesA* mutant could not be restored by *Klebsiella oxytoca nifQ* or *Escherichia coli moeB* completely. In addition, the *hesA* mutation does not affect nitrate reductase activity of *P. polymyxa* WLY78. Our results demonstrate *hesA* is a novel gene specially required for nitrogen fixation and its role is related to introduction of S and Mo into the FeMo-co of nitrogenase.

## INTRODUCTION

Nitrogenase, nitrate reductase, and a number of other enzymes require molybdenum and sulfur for their activity (*Johnson, 1980*; *Lester & Demoss, 1971*; *Pienkos, Shah & Brill, 1977*; *Shah et al., 1984*). Molybdenum and sulfur are found in these molybdoenzymes as parts of low-molecular-weight cofactors (*Johnson, 1980*; *Shah & Brill, 1977*). Two different molybdenum cofactors have been described: iron-molybdenum cofactor (FeMo-co), which is found only in nitrogenase (*Pienkos, Shah & Brill, 1977*; *Shah & Brill, 1977*), and molybdenum cofactor (Mo-co), which is found in other molybdoenzymes.

Most biological nitrogen fixation is catalyzed by molybdenum-dependent nitrogenase, which is distributed within bacteria and archaea. This nitrogenase is a two-component enzyme consisting of the Fe and MoFe proteins (*Hu & Ribbe, 2011*; *Rubio & Ludden, 2008*). The MoFe protein is an $\alpha_2\beta_2$ heterotetramer (encoded by *nifD* and *nifK*) that

Corresponding author
Sanfeng Chen, chensf@cau.edu.cn

contains two metalloclusters; FeMo-co, a [Mo-7Fe-9S-C-homocitrate] cluster which serves as the active site of substrate binding and reduction and the P-cluster, a [8Fe-7S] cluster which shuttles electrons to FeMo-co. The Fe protein (encoded by *nifH*) is a $\gamma_2$ homodimer bridged by an intersubunit [4Fe-4S] cluster that serves as the obligate electron donor to the MoFe protein (*Oldroyd & Dixon, 2014*). It is now well established from genetic and biochemical analysis that *nifE*, *nifN*, *nifX*, *nifB*, *nifQ*, *nifV*, *nifY* and *nifU*, *nifS* and *nifH* contribute to the synthesis and insertion of FeMo-co into nitrogenase (*Hu & Ribbe, 2011*; *Rubio & Ludden, 2008*). Of these genes, *nifQ* serves as a molybdenum donor for FeMo-co synthesis, *nifU* and *nifS* are involved in [Fe-S] cluster assembly. NifS is a cysteine desulfurase that provides sulfur for the assembly of transient [Fe-S] clusters onto the molecular scaffold NifU.

Mo-co consists of the molybdopterin cofactor with molybdenum bound to its dithiolene moiety. Mo-co is found in nitrate reductase of some fungi and bacteria, such as nitrate reductase of *Escherichia coli* (*Leimkühler, Wuebbens & Rajagopalan, 2011*; *Mendel, 2013*). In *E. coli*, several loci (*moa*, *mob*, *moe* and *mog*) are involved in the biosynthesis of the Mo-co (*Leimkühler, Wuebbens & Rajagopalan, 2001*). In Mo-co biosynthesis, MoeB functions to sulfurylate MoaD, and IscS functions as a primary sulfur-donating enzyme by interacting specifically with MoeB and MoaD in the biosynthesis of molybdopterin in *E. coli* (*Sambasivarao et al., 2002*; *Zhang et al., 2010*).

Diazotrophic (nitrogen-fixing) Gram-positive and endospore-formed *Paenibacillus* spp. have potential uses as a bacterial fertilizer in agriculture. Comparative genomic analysis shows that a *nif* gene cluster composed of nine genes *nifBHDKENXhesAnifV* is conserved in the genomes of *Paenibacillus polymyxa* WLY78 and other the diazotrophic *Paenibacillus* species and strains. Whereas, the Gram-negative diazotrophic model *Klebsiella oxytoca* has 20 *nif* genes (*nifJHDKTYENXUSVWZMFLABQ*) organized in seven transcriptional units and co-located within a 24 kb cluster (*Arnold et al., 1988*). Compared to *K. oxytoca*, *P. polymyxa* WLY78 has no *nifQ*, *nifS* and *nifU*, but contains a *hesA* gene. The *hesA* gene is also highly conserved within the *nif* gene clusters of Cyanobacteria and Actinobacteria (*Frankia*). The predicted product HesA is a member of the ThiF-MoeB-HesA family containing an N-terminal nucleotide binding domain and a C-terminal MoeZ/MoeB-like domain. However, the function of *hesA* gene in nitrogen fixation is unknown.

In this study, the function of the *P. polymyxa hesA* gene in nitrogen fixation is investigated. Mutation of *P. polymyxa hesA* leads to no $N_2$-fixation in the media containing low levels of molybdate. The effect of the mutation can be suppressed by the addition of high levels of molybdate or cystine. The *hesA* mutation does not affect nitrate reductase activity of *P. polymyxa* WLY78. Our results demonstrate *hesA* is specially required for nitrogen fixation.

## MATERIALS AND METHODS

### Bacterial strains and media

Bacterial strains used in this study are listed in Table S1. *P. polymyxa* WLY78, mutants and the recombinant *E. coli* strains were grown in LB medium at 37 °C. When appropriate, antibiotics for *E. coli* strains and *P. polymyxa* WLY78 mutants were required. For *P. polymyxa* WLY78 mutant strains, antibiotics were added at five mg/L

chloramphenicol (Cm), 12.5 mg/L tetracycline. While for *E. coli*, antibiotics were added at 100 mg/L ampicillin, 25 mg/L tetracycline and 25 mg/L Cm for maintenance of plasmids.

Nitrogen-limited medium (per liter contains: 10.4 g $Na_2HPO_4$, 3.4 g $KH_2PO_4$, 26 mg $CaCl_2·2H_2O$, 30 mg $MgSO_4$, 0.3 mg $MnSO_4$, 36 mg Ferric citrate, 7.6 mg $Na_2MoO_4·2H_2O$, 10 μg p-aminobenzoic acid, 10 μg biotin, four g glucose as carbon source and 0.3 g glutamic acid as nitrogen source) was used for assay of nitrogenase activity (*Wang et al., 2013*). When necessary, different concentrations of $Na_2MoO_4$ or cystine or sulfate were supplemented in the nitrogen-limited medium.

To determine nitrate reductase activity, *P. polymyxa* WLY78 and mutant strains were cultured in mineral salts medium (per liter contains: 9.16 g $K_2HPO_4·3H_2O$, 1.0 g $(NH_4)_2SO_4$, 0.22 g $MgSO_4·7H_2O$, 2.0 g $KH_2PO_4$, 8.0 g nutrient broth, 10 g $KNO_3$, 10 g glucose) (*Sperl & Demoss, 1975*).

## Enzyme assays

For nitrogenase activity assays, *P. polymyxa* WLY78, mutants, the complemented strains and the recombinant *E. coli* strains were grown in 20 mL of LB media (Supplemented with antibiotics) in 50 mL flasks shaken at 200 rpm overnight at 37 °C. The cultures were collected by centrifugation, precipitations were washed three times with sterilized water and then resuspended in nitrogen-limited medium containing different concentrations of $Na_2MoO_4$ (supplemented with antibiotics for mutants and the engineered *E. coli* strains when necessary) to a final $OD_{600\ nm}$ of 0.2–0.5. The nitrogenase activity was determined by the procedure described by *Xie et al. (2012)*.

For the assay of nitrate reductase, *P. polymyxa* WLY78 and mutant strains were grown in 50 mL of LB media in 100 mL flasks shaken at 200 rpm for 16 h at 37 °C. The cultures were collected by centrifugation, washed three times with sterilized water and then resuspended in mineral salts medium. The cultures were grown at 200 rpm at 37 °C anaerobically. The nitrate reductase was determined by the method described by Garzón (*Garzón et al., 1992*). Nitrate reductase activity was expressed as nanomoles of nitrite·minute$^{-1}$·milligram$^{-1}$ of cell protein.

## Growth of the wild type and mutant strains

To measure growth of strains cultured in different concentrations of $Na_2MoO_4$, cells were grown in 20 mL of LB media in 50 mL flasks shaken at 200 rpm at 37 °C overnight. The cultures were collected by centrifugation, washed three times with sterilized water and then resuspended in sufficient nitrogen medium (nitrogen-limited medium supplemented with 100 mM $NH_4Cl$) containing different concentrations of $Na_2MoO_4$ to a start $OD_{600\ nm}$ of 0.1. After incubating the cultures for 48 h at 37 °C with shaking at 250 rpm, the maximum growth of *P. polymyxa* WLY78 and mutant strains were determined by absorbancy at 600 nm.

## Construction of *Paenibacillus hesA* mutant strain

The *Paenibacillus hesA* mutant was constructed through homologous recombination with pRN5101 (*Hou et al., 2016*), which carries a temperature-sensitive region from plasmid

pE194ts (*Villafane et al., 1987*) containing two replication origins: one is pMB1 which can be reproduced in *E. coli* and the other replicon is from a plasmid pE194 which can be reproduced in Gram-positive *Bacillus* (*Gryczan, Contente & Dubnau, 1978*). A 3432 bp DNA fragment, including a 1,119 bp upstream sequence and a 1,116 bp downstream fragment of the *hesA* coding region, was cloned to pRN5101. Then, a 1,197 bp DNA fragment, including the promoter and coding region of Cm resistant gene, was amplified from plasmid pPR9TT and inserted between upstream and downstream homologous arms in the opposite direction from the *nif* genes, generating plasmid pRN5101-Cm. The plasmid pRN5101-Cm was introduced into *P. polymyxa* WLY78 by electroporation (BTX, ECM 399) for homologous recombination. Transformants resistant to Cm were grown at 30 °C and transferred to five mL of LB media (supplemented with five mg/L Cm) in 20 mL test tubes shaken at 200 rpm for 36 h at 39 °C for about five generations. The strains were then plated onto LB agar plates supplemented with Cm and incubated for 24 h at 39 °C. Integration of the recombinant plasmid was confirmed by PCR and sequencing. The deletion mutant strain was designated as Δ*hesA*.

## Construction of recombinant plasmids for genetic complementation assays

The genes of *hesA*, *nifQ* and *moeB* were PCR amplified from genomic DNA of *P. polymyxa* WLY78, *K. oxytoca* M5a1 and *E. coli* K12, respectively. These PCR products, being under the control of *nifB* promoter of *P. polymyxa* WLY78, were cloned to the plasmid pHY300PLK, respectively. Then the recombinant plasmids were introduced into *P. polymyxa hesA* mutant, respectively.

For complementation of the *hesA* deletion of the recombinant *E. coli* 78-7 which carries a *nif* gene cluster composed of nine genes (*nifBHDKENXhesAnifV*) from *P. polymyxa* WLY78 (*Wang et al., 2013*), the *P. polymyxa hesA* coding region, being under the control of *nifB* promoter of *P. polymyxa* WLY78 was cloned to vector pBluescript II SK (+) that contained a Cm expression cassette, producing the recombinant plasmid pBluescript II SK (+)-*hesA*. Primers for PCR are listed in Table S2. Recombinant plasmids and strains are listed in Table S3.

## Quantitative real-time PCR analysis of *nifV* expression

For quantitative real-time-PCR (qRT-PCR), cultures of *P. polymyxa* WLY78 and Δ*hesA* were grown in $N_2$-fixing conditions (two mM glutamate and without $O_2$) and harvested after 20 h of incubation. Total RNA was isolated using TRIzol (cat. no. 9108; Takara Bio, Dalian, China) (*Rio et al., 2010*). The possibility of contamination of genomic DNA was eliminated by digestion with RNase-free DNase I (cat. no. RR047A; Takara Bio, Dalian, China). The integrity and size distribution of the RNA were verified by agarose gel electrophoresis, and the concentrations were determined spectrophotometrically. Synthesis of cDNA was carried out using RT Prime Mix according to the manufacturer's specifications (cat. no. RR047A; Takara Bio, Dalian, China). 0.4 μg of cDNA was used for qRT-PCR. The relative transcript levels of *nifV* were determined with 16S rDNA as a

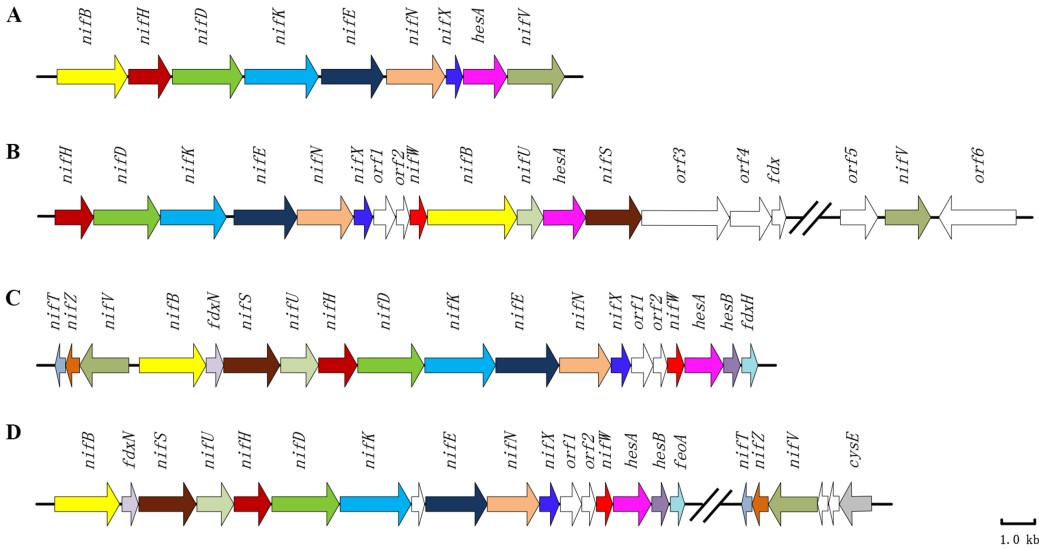

**Figure 1 Clusters of nitrogen-fixation related genes between *P. polymyxa* WLY78 and other nitrogen-fixing bacteria.** (A) *P. polymyxa* WLY78, (B) *Frankia* sp. EAN1pec, (C) *Cyanothece* sp. ATCC 51142, and (D) *Anabaena* sp. PCC 7120.

control by SYBR Premix Ex Taq (Tli RNaseH Plus) kit (cat. no. RR820A; Takara Bio). Primers for *nifV* and 16S rDNA used for qRT-PCR are listed in Table S2.

## Statistical analysis

Statistical analyses were performed using SPSS 20.0 (SPSS, Chicago, IL, USA). The least significant test was calculated at the 0.05 or 0.01 probability level for different treatment mean comparisons of nitrogenase activity.

## RESULTS

### A *hesA* gene is highly conserved in *nif* gene clusters among diazotrophic *Paenibacillus,* cyanobacteria and *Frankia*

*Paenibacillus polymyxa* WLY78 possesses a minimal *nif* gene cluster consisting of nine genes (*nifB nifH nifD nifK nifE nifN nifX hesA* and *nifV*). Our previous results demonstrated that the recombinant *E. coli* 78-7 which was generated by introducing the *P. polymyxa nif* gene cluster into *E. coli* JM109 could produce active nitrogenase (*Wang et al., 2013*). Notably, a *hesA* gene is contained within the *nif* gene cluster, and its predicted product HesA is a member of the ThiF-MoeB-HesA family containing an N-terminal nucleotide binding domain and a C-terminal MoeZ/MoeB-like domain. NCBI blast analyses reveal that *hesA* gene is significantly conserved within *nif* gene clusters among diazotrophic *Paenibacillus,* cyanobacteria and *Frankia* (Fig. 1). HesA from *P. polymyxa* WLY78 at amino acid sequence level shares more than 90% identity with those of other diazotrophic *Paenibacillus* spp. (*Xie et al., 2014*) and 46.8%, 43.1% and 49.2% identity with HesA of *Frankia alni* ACN14a (*Chang et al., 2012*), *Cyanothece* sp. ATCC 51142 (*Welsh et al., 2008*) and *Anabaena* sp. PCC 7120 (*Mitschke et al., 2011*), respectively.
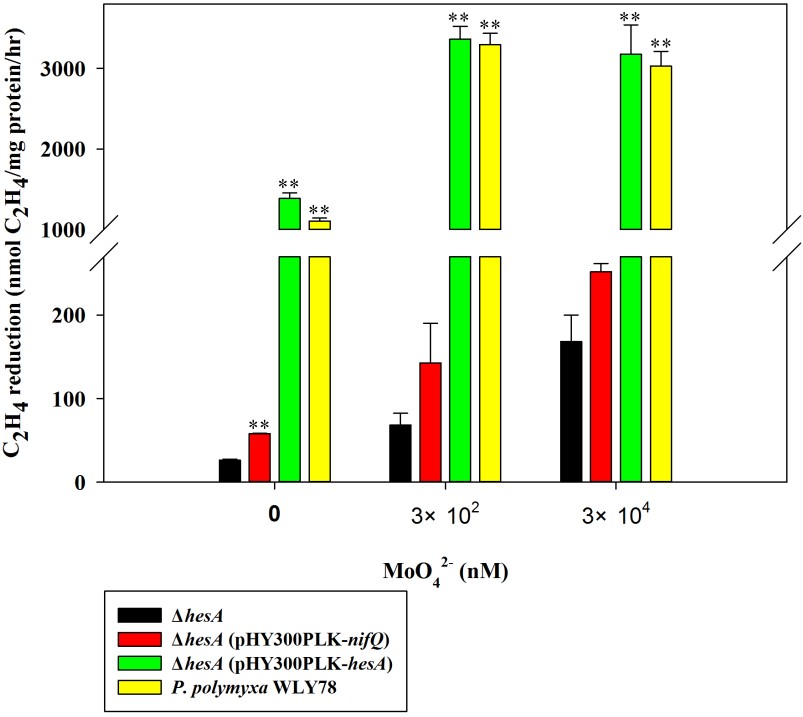

**Figure 2** Relative nitrogenase activity of *P. polymyxa* WLY78, the Δ*hesA* mutant, the complemented strains Δ*hesA* (pHY300PLK-*hesA*) and Δ*hesA* (pHY300PLK-*nifQ*). *x*-Axis indicates the concentration of molybdate added to the culture medium with 0.25 mM sulfate as sulfur source. The nitrogenase activity of the Δ*hesA* mutant was used as a control. Date represent the mean ± SD (*n* = 3). ** Indicates significant differences between control and other treatments determined by LSD at *P* < 0.01.

Furthermore, we find that *P. polymyxa* HesA has 32%, 30%, 26% and 24% identity with MoeB of *Pseudomonas fluorescens, K. oxytoca, E. coli* and *Bacillus subtilis*, respectively, whereas it shows only 9% identities with NifQ of *K. oxytoca* and *A. vinelandii* and 8–18% with these NifQs from other N₂-fixing bacteria. The data indicate that HesA exhibits higher similarity to MoeB than to NifQ. Also, there are no *moeB*-like genes other than *hesA* in *Paenibacillus*.

## The *hesA* mutation of *P. polymyxa* WLY78 leads to nearly complete loss of nitrogenase activity

As described in Materials and Methods, the Δ*hesA* mutant was constructed by inserting a 1,197 bp Cm resistance gene cassette into the *nif* gene cluster (*nifBHDKENXhesAnifV*) and substituting for *hesA*. qRT-PCR further revealed that the mRNA level of *nifV* gene was not affected by the *hesA* mutation (Fig. S1), suggesting that the insertion of Cm-cassette into *nif* gene cluster did not generate polarity effect.

Here, the nitrogenase activities of the Δ*hesA* mutant and wild-type *P. polymyxa* WLY78 grown in medium containing 0.25 mM sulfate as sulfur source and supplemented with 0.3 and 30 μM molybdate, respectively, were comparely analyzed (Fig. 2). The Δ*hesA* mutant showed a significant decrease in nitrogenase activity compared to wild type, suggesting that *hesA* is involved in nitrogen fixation. We noticed that the *hesA* mutant and

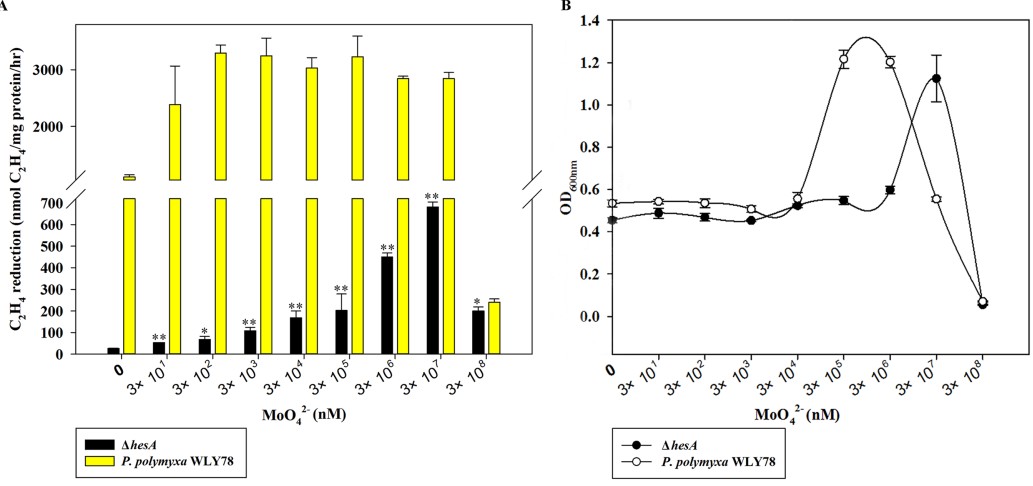

**Figure 3 Effect of the molybdate concentration on nitrogenase activity (A) and maximum growth (B) of *P. polymyxa* WLY78 and the Δ*hesA* mutant.** *x*-Axis indicates the concentration of molybdate added to the culture medium with 0.25 mM sulfate as sulfur source. The maximum growth means culture absorbance after 48 h under diazotrophic growth conditions. The nitrogenase activity of the Δ*hesA* mutant in the absence of added sodium molybdate was used as a control. Date represent the mean ± SD (*n* = 3). * or ** indicate significant differences between control and other treatments determined by LSD at *P* < 0.05 or *P* < 0.01.

*P. polymyxa* WLY78 also exibited some activity in medium without molybdate, which might be due to trace molybdenum in the medium.

Furthermore, we demonstrated that *P. polymyxa hesA* could completely restore the activity of *hesA* mutant to the level of wild-type *Paenibacillus,* confirming that *hesA* is required for nitrogen fixation. We also tried to complement with *K. oxytoca nifQ* and found that the *nifQ* could not suppress the effect of *hesA* mutation completely (Fig. 2). The data suggest that *K. oxytoca nifQ* could not act as a molybdenum donor for FeMo-co synthesis of nitrogenase in *P. polymyxa.*

## The *hesA* mutant requires high levels of molybdate for nitrogenase activity

The molybdate requirement for nitrogenase activity in the Δ*hesA* mutant was studied, with *P. polymyxa* WLY78 as a control. As shown in Fig. 3A, *P. polymyxa* WLY78 had nitrogenase activity even in the absence of added sodium molybdate, because there were always traces of Mo in other chemicals, the glassware and water. The maximum activity of wild-type *P. polymyxa* WLY78 was obtained at 0.3 μM molybdate, and the nitrogenase activity did not increase as the concentration of molybdate increased. In contrast, the nitrogenase activity of the Δ*hesA* mutant improved as the molybdate concentration increased, and the maximum activity was obtained at 30 mM molybdate. The maximum activity of the Δ*hesA* mutant at 30 mM molybdate was 25% of that of the wild type obtained at 0.3 μM molybdate, indicating that the addition of very high levels of molybdate could partially suppress the effect of the mutation.

Diazotrophic growth of the wild-type *P. polymyxa* WLY78 and Δ*hesA* mutant showed the same dependence on the molybdate added (Fig. 3B). Compared with the wild type,

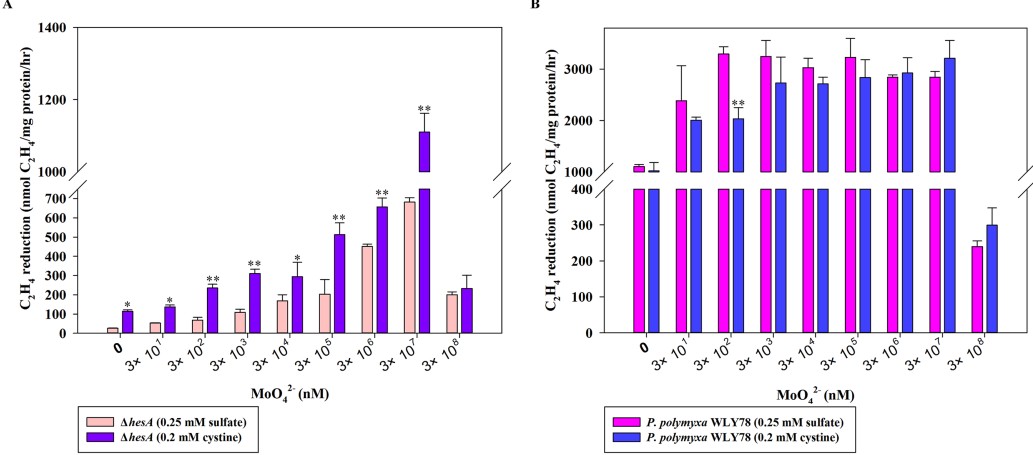

**Figure 4 Effect of the sulfur sources on the nitrogenase activity of the ∆hesA mutant and P. polymyxa WLY78.** (A) Nitrogenase activity of ∆hesA with two sulfur sources. (B) Nitrogenase activity of P. polymyxa WLY78 with two sulfur sources. x-Axis indicates the concentration of molybdate added to the culture medium with 0.25 mM sulfate or 0.2 mM cystine or the combination of sulfate and cystine as sulfur source. Date represent the mean ± SD ($n = 3$). * or ** indicate significant differences among different treatments determined by LSD at $P < 0.05$ or $P < 0.01$.

the ∆hesA mutant needed higher concentration of molybdate for the maximum growth. Millimolar concentrations of molybdate obviously inhibited both activity and diazotrophic growth. Meanwhile, the ∆hesA mutant showed a higher tolerance.

## Effect of two sulfur sources on nitrogenase activity of the ∆hesA mutant

Here, the effects of two sulfur sources on nitrogenase activity of the ∆hesA mutant were determined. As shown in Fig. 4A, the nitrogenase activity of the ∆hesA mutant in medium containing cystine as the only sulfur source was two to five times higher than that in medium containing sulfate at all of different concentrations of molybdenum except for a high inhibitory level of 0.3M molybdenum. However, no difference in nitrogenase activity of wild-type Paenibacillus was observed when using either sulfate or cystine as sulfur source (Fig. 4B). These results demonstrated that the use of cystine as the only sulfur source could suppress the effect of hesA mutation on nitrogenase activity, but the addition of cystine could not affect the requirement of molybdenum for maximum nitrogenase activity. The results indicated that hesA might be involved in sulfur transport.

## High levels of cystine restore the nitrogenase activity of the ∆hesA mutant, but E. coli moeB cannot

As shown in Fig. 5A, the nitrogenase activity of the ∆hesA mutant was improved with the addition of cystine, and higher concentration of cystine was required for the ∆hesA mutant to reach the maximum activity of the wild type. Similarly, the activity of the ∆hesA mutant increased with the addition of sulfate, but a higher concentration (above 1.0 mM) of sulfate was required for the ∆hesA mutant to reach the maximum activity of the wild type (Fig. 5B).

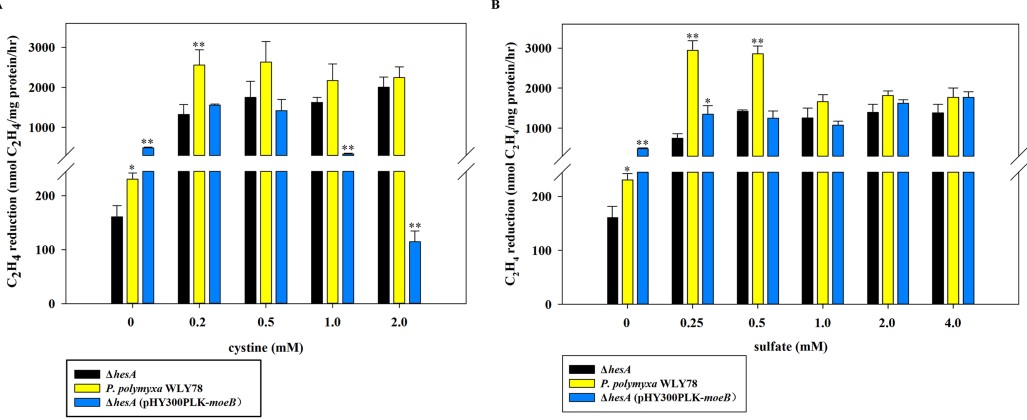

**Figure 5 Effect of the different concentrations of cystine (A) or sulfate (B) on nitrogenase activity of ΔhesA, *P. polymyxa* WLY78 and the complemented strain ΔhesA (pHY300PLK-*moeB*).** (A) Nitrogenase activity of the strains with different cystine concentrations. (B) Nitrogenase activity of the strains with different sulfate concentrations. *x*-Axis indicates the concentration of cystine or sulfate added to the culture medium with 30 mM molybdate. The nitrogenase activity of the ΔhesA mutant was used as a control. Date represent the mean ± SD (*n* = 3). * or ** indicate significant differences between control and other treatments determined by LSD at *P* < 0.05 or *P* < 0.01.

In addition, we tried to complement the ΔhesA mutant with *E. coli moeB* and found that the expression of *moeB* in the mutant did not improve nitrogenase activity (Fig. 5). In fact, the nitrogenase activity in the ΔhesA mutant expressing *moeB* declined when the concentration of cystine exceeded 1.0 mM. The data suggested that *E. coli moeB* could not completely suppress the effect of the *hesA* mutation when sulfate or cystine was used as the only sulfur source. It has been reported that MoeB, MPT synthase sulfurase, is involved in sulfuration of MPT synthase (*Leimkühler, Wuebbens & Rajagopalan, 2001*; *Pitterle & Rajagopalan, 1993*; *Rajagopalan, 1997*).

## A high level of molybdate is required for nitrogenase activity of the recombinant *E. coli* 78-7 with a *hesA* deletion

In order to confirm the role of the *hesA* in nitrogen fixation, we compared the activities of the recombinant *E. coli* 78-7 strain carrying a *nif* gene cluster (*nifBHDKENXhesAnifV*) of *P. polymyxa* WLY78 (*Wang et al., 2013*) and the *hesA* deletion strain (D-O), which was a derivative of *E. coli* 78-7 strain without the *hesA* gene. As shown in Fig. 6, the nitrogenase activity of the D-O strain was improved as the concentration of molybdate increased. The D-O strain required 10 times as much molybdate to achieve the level of activity of the *E. coli* 78-7 strain. The *P. polymyxa hesA* gene could complement the D-O strain and restore the activity of the *hesA* deletion strain to that of the *E. coli* 78-7 strain, suggesting that the addition of very high levels of molybdate can suppress the effect of the *hesA* mutation. The current result was consistent with the results obtained in the *P. polymyxa hesA* mutant.

## Cystine can enhance activity of the *hesA* deletion in *E. coli* 78-7

Here, we further investigated the effect of cystine on the activities of the *hesA* deletion strain (D-O) which carried other *nif* genes (*nifBHDKENXV*) without *hesA* when it was

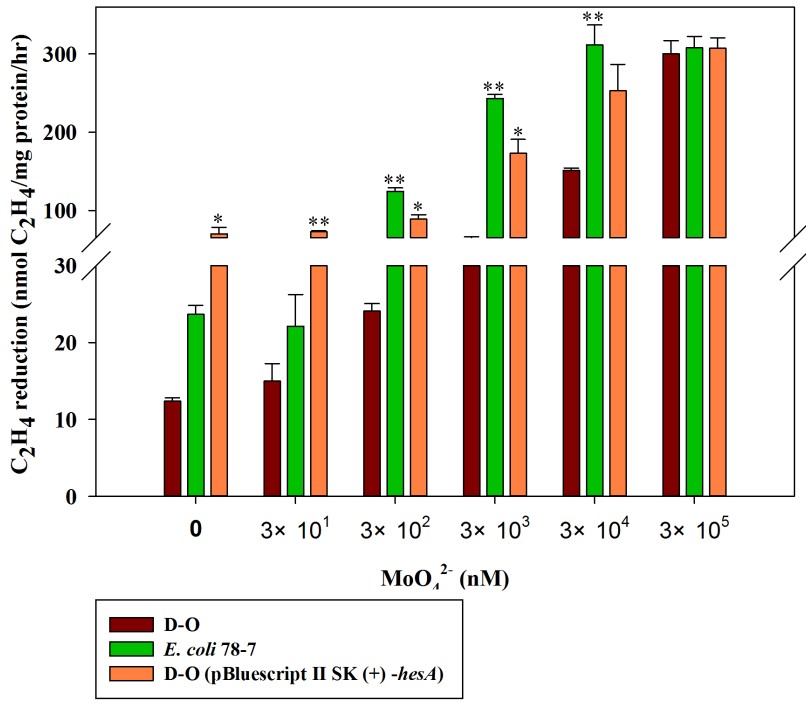

**Figure 6 Effect of the molybdate concentration on nitrogenase activity of the recombinant _E. coli_ 78-7, the _hesA_ deletion strain (D-O) and the complemented strain D-O (pBluescript II SK (+)-_hesA_).** _x_-Axis indicates the concentration of molybdate added to the culture medium with 0.25 mM sulfate as sulfur source. The nitrogenase activity of the D-O strain was used as a control. Date represent the mean ± SD (_n_ = 3). * or ** indicate significant differences between control and other treatments determined by LSD at _P_ < 0.05 or _P_ < 0.01.

grown in medium supplemented with 30 μM molybdate and with different concentrations of cystine. As shown in Fig. 7, cystine could enhance activity of the _hesA_ deletion strain (D-O), but it did not have the similar effects on the complemented strain D-O (pBluescript II SK (+)-_hesA_).

## The _P. polymyxa hesA_ mutation does not affect nitrate reductase activity

Nitrate reductase, like nitrogenase, has been characterized as a molybdoenzyme (_Taniguchi & Itagaki, 1960_) and molybdenum is necessary for nitrate reductase (_Lester & Demoss, 1971_). Here, the nitrate reductase activities of the Δ_hesA_ mutant and wild-type _P. polymyxa_ WLY78 were comparatively analyzed. The Δ_hesA_ mutant (1,048.12 ± 28.80 nmol nitrite·min$^{-1}$·mg$^{-1}$protein) and wild-type _P. polymyxa_ WLY78 (1,057.37 ± 5.95 nmol nitrite·min$^{-1}$·mg$^{-1}$protein) showed the similar nitrate reductase activity, indicating that the _hesA_ gene product is required for synthesis of FeMo-co, but not implicated in the biosynthesis of Mo-co.

## DISCUSSION

A _nif_ gene cluster composed of nine genes _nifBHDKENXhesAnifV_ is conserved in the genomes of _P. polymyxa_ WLY78 and other the diazotrophic _Paenibacillus_ spp.

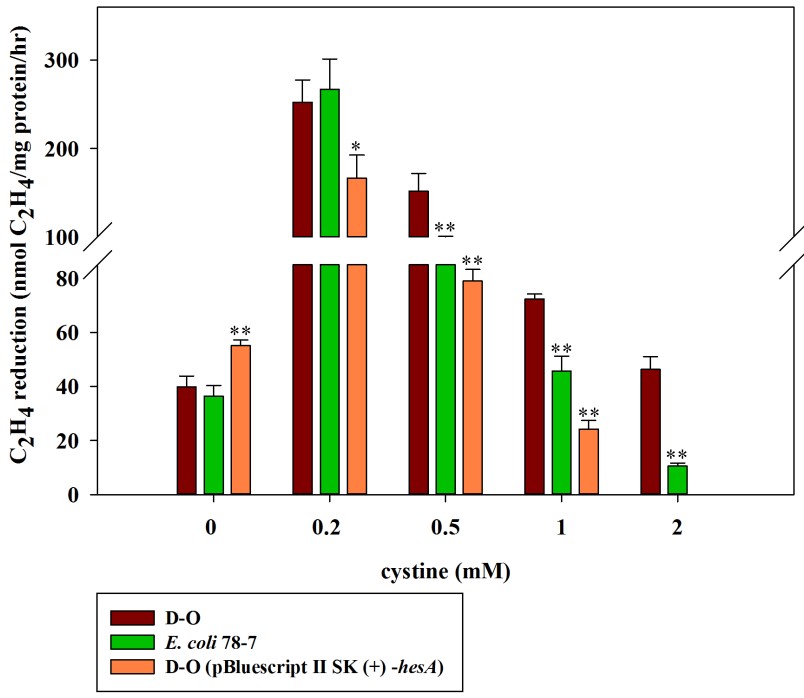

**Figure 7 Effect of the cystine concentration on nitrogenase activity of the recombinant *E. coli* 78-7, the *hesA* deletion strain (D-O) and the complemented strain D-O (pBluescript II SK (+)- *hesA*).** *x*-Axis indicates the concentration of cystine added to the culture medium with 30 μM molybdate. The nitrogenase activity of the D-O strain was used as a control. Date represent the mean ± SD (*n* = 3). * or ** indicate significant differences between control and other treatments determined by LSD at *P* < 0.05 or *P* < 0.01.

We find that the *nif* gene cluster has a *hesA*, but no *nifQ* and *nifSU*. The genome of *P. polymyxa* WLY78 has molybdate transport genes, including *modA*, *modC* and *modF*. It also contains *moaA*, *moaB*, *moaC*, *moaD*, *moaE*, *mobA*, *mobB* and *moeA* except for *moeB*, and contains a complete *suf* (*sufCBSUD*) operon, a partial *suf* (*sufABC*) operon, a partial *isc* system (*iscSR* and *fdx*) and two *nifS*-like genes (*Shi et al., 2016*).

The function of *hesA* gene of diazotrophic *Paenibacillus* is here for the first time to be investigated. Mutation of *P. polymyxa hesA* leads to nearly complete loss of nitrogenase activity. Similar result was reported that loss of *hesA* resulted in about a 50% loss of nitrogenase in Anabaena (*Borthakur et al., 1990*). Furthermore, we demonstrated that the effect of the mutation could be suppressed by the addition of high levels of molybdate to the medium. We also showed that deletion of *hesA* from the *nif* gene cluster made the recombinant *E. coli* 78-7 require high concentration of molybdate for nitrogenase activity. Our complementation studies demonstrated that *P. polymyxa hesA* could restore the nitrogenase activity of *hesA* mutant to that of wild-type *Paenibacillus* or that of the recombinant *E. coli* 78-7, but *K. oxytoca nifQ* could not, indicating that *K. oxytoca* NifQ cannot replace HesA in the original host. The results indicate that *P. polymyxa hesA* gene plays a role in molybdenum transport to nitrogenase. Similar results that NifQ⁻ and Mol⁻ mutants of *K. oxytoca* required high concentration of molybdate for nitrogenase activity were reported (*Imperial et al., 1984*, *1985*). There were

no functional differences among the NifQ⁻ and Mol⁻ mutants of *K. oxytoca* with respect to FeMo protein when molybdate was in excess (*Imperial et al., 1985*). Our results and their results suggest that NifQ is not necessary when molybdate is in excess. It was reported that the FeMo-co of nitrogenase could be completely synthesized in vitro by $Fe^{2+}$, $S^{2-}$, $MoO_4^{2-}$ and *R*-homocitrate, using purified NifB, NifEN and NifH proteins (*Curatti et al., 2007*). Curatti et al.'s reports suggested that NifQ is not necessary when molybdate is in excess in vitro synthesis of FeMo-co. One possible explanation is that NifQ cannot act as Mo donor in *Paenibacillus*.

Sulfur is an essential element for FeMo-co of nitrogenase and Mo-co of other enzymes and it can be obtained from various compounds. Generally, sulfate ($SO_4^{2-}$) and thiosulfate ($S_2O_3^{2-}$) are the preferred sulfur sources for the majority of $N_2$-fixing organisms (*Aguilar-Barajas et al., 2011*). It was shown that the physiological sulfur donor for the formation of the dithiolene group of molybdopterin in Mo-co biosynthesis was likely to be *L*-cysteine and IscS functioned as a primary sulfur-donating enzyme by interacting specifically with MoeB and MoaD in the biosynthesis of molybdopterin in *E. coli* (*Sambasivarao et al., 2002*; *Zhang et al., 2010*). MoeB contains several conserved cysteine residues, one of which has been postulated to be involved in the formation of a thioester linkage between MoaD and MoeB (*Rogers, Crescenzo & Söll, 1995*).

In this study, we demonstrated that cystine did not modify the requirement of molybdate for nitrogenase activity in the wild-type *P. polymyxa* WLY78. Whereas, the nitrogenase activity of the Δ*hesA* mutant was higher when it was grown in medium containing cystine as the only sulfur source than sulfate as the only sulfur source. Moreover, cystine could enhance activity of *hesA* deletion of *E. coli* 78-7. However, *P. polymyxa hesA* gene could not complement the *hesA* deletion of *E. coli* 78-7 when cystine was high in the medium. We do not know whether *E. coli moeB* plays a role in nitrogen fixation of the *hesA* mutant from the recombinant *E. coli* 78-7. It was reported that the effects of *nifQ* and *mol* mutations of *K. oxytoca* on nitrogenase could be suppressed by the addition of cystine, suggesting that a sulfur donor and molybdenum interact at an early step in the biosynthesis of the iron-molybdenum cofactor (*Dos Santos & Dean, 2008*). Sulfate interferes with the utilization of molybdate, which becomes apparently only when the *nifQ* or *mol* products are not active in *K. oxytoca* (*Dos Santos & Dean, 2008*). A competition between sulfate and molybdate was found in *Clostridium pasteurianum* (*Elliott & Mortenson, 1975*) and *Aspergillus niger* (*Tweedie & Segel, 1970*) at the level of transport into the cell. The transport of sulfate and molybdate in *K. oxytoca* are independent and specific, and that the sulfur source effect on molybdate requirement is intracellular.

Furthermore, we comparatively analyzed the nitrate reductase activity of the Δ*hesA* mutant and wild-type *P. polymyxa* WLY78, and our results showed that the *hesA* mutation did not affect nitrate reductase activity. The results confirm that the *P. polymyxa hesA* is involved in FeMo-co synthesis, but not in Mo-co synthesis.

## ACKNOWLEDGEMENTS

We would like to thank associate professor Yanqin Ding from Shandong Agricultural University for proving the plasmid pRN5101.

### Funding

This work was supported by the Innovative Project of SKLAB (Grant No. 2017SKLAB1-1) and the Ministry of Education Basic Research Business Expenses (Grant No. 2016SY001). The funders had no role in study design, data collection and analysis, decision to publish, or preparation of the manuscript.

### Grant Disclosures

The following grant information was disclosed by the authors:
Innovative Project of SKLAB: 2017SKLAB1-1.
Ministry of Education Basic Research Business Expenses: 2016SY001.

### Competing Interests

The authors declare that they have no competing interests.

### Author Contributions

- Xiaomeng Liu conceived and designed the experiments, performed the experiments, analyzed the data, prepared figures and/or tables, authored or reviewed drafts of the paper, approved the final draft.
- Xiyun Zhao contributed reagents/materials/analysis tools.
- Xiaohan Li contributed reagents/materials/analysis tools.
- Sanfeng Chen conceived and designed the experiments, authored or reviewed drafts of the paper, approved the final draft.

### Data Availability

   The raw measurements are available as Supplemental Files.

### Supplemental Information

Supplemental information for this article can be found online at http://dx.doi.org/10.7717/peerj.6294#supplemental-information.

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
