# Peer review of "Suppression of hesA mutation on nitrogenase activity in Paenibacillus polymyxa WLY78 with the addition of high levels of molybdate or cystine"

_PeerJ, doi:10.7717/peerj.6294_

## Round 0.1 · original submission · Major Revisions

Both reviewers agree on the importance of your manuscript and highlight several instances where the text can be improved. Pay special attention to reviewer#1's comments regarding the behavior of all strains towards cystine, as no major differences are apparent between DeltahesA and other strains.

·

Basic reporting

The manuscript describes the effect of the disruption of hesA, which is part of the nitrogen fixation gene cluster of the bacterium Paenibacullus polymyxa, on nitrogenase activity. The authors show that the effect of the hesA disruption can be partially rescued by molybdate and sulfur source supplementation. The work is scientifically sound and the manuscript provides a complete description of methods and results. The findings are new and interesting, yet the manuscript can be significantly improved.

1. major comment: the introduction lacks sufficient introduction of the FeMoco biosynthesis pathway. The gene cluster is briefly mention, but existing knowledge about the role and interaction of the different components should be properly introduced. I have a similar comment about the discussion section. The new findings should be placed in context of the existing knowledge on the biosynthesis of the nitrogenase metal cofactors. The focus is now too much on only HesA and related proteins, not on the interplay with the other components of the FeMoco.
2. the text still contains a number of typographical errors, to list a few:
line 253: concentraion
line 281: excecpt
line 288: Furthwemore
line 307: genrally
line 338: affact
3. throughout the manuscript the authors express the molybdate concentration in nM, ranging from 3E1 to 3E8 nM. I strongly recommend to use suitable units: e.g. 30 nM instead of 3E1 nM, and 0.3 M instead of 3E8 nM. This will assist the reader in understanding the actual concentrations that have been used.

Minor comments below:
4. abstract: "will provide insight into the complex synthesis..". The authors can be more specific here. How does this discovery changed our current understanding of FeMoco biosynthesis?
5. line 60: "molybdenum cofactor (Mo-co)". It is worthwhile to introduce already here that Moco consists of the molybdopterin cofactor with molybdenum bound to its dithiolene moiety.
6. line 260: cysteine enhances the activity of the hesA deletion in the E. coli train containing the nitrogen fixation gene cluster up to a level which is similar to the activity with sulfate as a sulfur donor (see fig 6 and 7) at the same Mo level. I do not see this clearly described in the text.
7. line 266-268 and fig 7: all strains exhibit similar behaviour in response to the cystine concentration, with or without HesA. I am not convinced that this "confirms that hesA might be involved in sulfur transport like the function of MoeB".
8. line 316-319: I am not convinced that nitrogenase of the hesA mutant is so much higher with cysteine than with sulfate. In Fig 5 the scales and the color scheme of the bars are unfortunately different, but if I see correctly
9. line 319: what do the authors mean exactly that HesA is involved in cystine transportation? Does it reduce cysteine to cysteine?
10. discussion: I miss a proposed mechanism as part of the discussion. Clearly HesA is involved in S and Mo introductin into the FeMoco. Is it possible to place HesA functionally between the other components of the cofactor biosynthesis system that is present in this organism?
11: line 329: the authors comment on the presence of ModA(B)C. For most of the Mo concentrations that have been used active transport is likely not necessary. Furthermore, if the intercellular concentration of Mo is high, the transport system is likely not expressed.
12. line 337-340: It is interesting that HesA is apparently not involved in Moco biosynthesis, since this organism does not have MoeB (if I understood that correctly). Can the authors explain why this is not a problem for Moco biosynthesis?

Experimental design

The research question is clear and the experiments have been well designed and adequately described. It would be very interesting to measure the action of the HesA protein in-vitro, but this can be follow-up research.

Validity of the findings

The results are clear and the interpretation is overall sound. My main comment, as stated under (1) is that cystine and sulfate appear to have a similar effect on the nitrogenase activity. The authors report that cystine strongly enhances the nitrogenase activity of the hesA mutant, but this does not clearly follow from the data.

Reviewer 2 ·

Basic reporting

The article is written in mostly good English and uses clear, unambiguous, technically correct language. It does conform to professional standards of courtesy and expression. It has sufficient introduction and background to demonstrate how the work fits into the broader field of knowledge and relevant prior literature is appropriately referenced. The structure of the article is in a format of ‘standard sections’. This manuscript is ‘self-contained,’ represents an appropriate ‘unit of publication’, and includes results relevant to the hypothesis.

Experimental design

This manuscript provides evidence that hesA in P. polymyxa is involved in the pathway from Mo and S transport to synthesis of Fe-Mo-co. The experiments are well designed and the question is interesting.. The data in this manuscript provide evidence that in both P. polymyxa and in E. coli expressing nif genes, loss of nitrogenase activity in a hesA mutant can be partly overcome by increasing levels of Mo or cysteine, but not sulfate. The methods for nitrogenase activity are inadequate. While details are not necessary, the fact that it was apparently done by measuring acetylene reducton to ethylene by GC should be indicated.

Validity of the findings

The data in this manuscript are robust, statistically sound and controlled. and they provide evidence that in both P. polymyxa and in E. coli expressing nif genes, loss of nitrogenase activity in a hesA mutant can be partly overcome by increasing levels of Mo or cystine, but not sulfate. The fact that a hesA mutant has no effect on nitrate reductase that uses Mo-co indicates that hesA is specifically involved in FeMo-co biosynthesis. While the data generally support the conclusions there is nothing in this manuscript that provides any information on how HesA is involved in synthesis of Fe-Mo-co.

Additional comments

Fig. 1 provides no information on the degree of similarity of the hesA genes. Perhaps the authors could create a table showing % similarly or provide a phylogenetic tree.

Fig 8 is unnecessary. The numbers should simply be reported in the text.

Line 133. “Erythromycin” should be “chloramphenicol.”

Line 195. “Respectatively” is misspelled

Lines 236 and 238 change “than” to “of”.

Lines 240 and 241. Complemented is not the correct word when a different gene is used. Here is better wording: “In addition, we tried to complement the ΔhesA mutant with E. coli moeB and found that the expression of moeB in the mutant did not improve nitrogenase activity (Fig. 5). In fact, the nitrogenase activity in the ΔhesA mutant expressing moeB declined when the concentrate of cystine exceeded 1.0 mM.“

Line 253. Rewrite as follows: “The D-O strain required 10 times as much molybdate to achieve the level of activity of the E. coli 78-7 strain. The P. polymyxa hesA gene could complement the D-O strain and restore the activity of the hesA deletion strain to that of the E. coli 78-7 strain.

Line 256. The meaning of this sentence is not clear. I do not understand what “restrain” means here.

Line 288. I do not think that the Mitschke et al. 2011 reference provides any direct information that hesA is required for nitrogenase. In fact Borthakur et al. 1990 indicates that loss of hesA results in only about a 50% loss of nitrogenase in Anabaena.

---

## Round 0.2 · accepted · Accept

Thank you for addressing the reviewer's comments.

# Reviewer 2 ·

Basic reporting

Acceptable.

Experimental design

Acceptable.

Validity of the findings

Valid.

Additional comments

An improved manuscript that adequately addresses most of the concerns of the reviewers.